# Variation of the Main Alkaloid Content in *Equisetum palustre* L. in the Light of Its Ontogeny

**DOI:** 10.3390/toxins12110710

**Published:** 2020-11-09

**Authors:** Jürgen Müller, Philipp Mario Puttich, Till Beuerle

**Affiliations:** 1Group Grassland and Forage Science, University of Rostock, Justus-von-Liebig-Weg 6, 18159 Rostock, Germany; 2Institute of Pharmaceutical Biology, Technical University of Braunschweig, Mendelssohnstr. 1, 38106 Braunschweig, Germany; p.puttich@tu-braunschweig.de (P.M.P.); t.beuerle@tu-bs.de (T.B.)

**Keywords:** marsh horsetail, palustrine, palustridiene, forage toxicity, livestock poisoning, wet grassland management

## Abstract

Marsh horsetail (*Equisetum palustre* L.) is one of the most poisonous plants of wet grasslands in the northern hemisphere, which poses a major health threat to livestock. Available data on the levels of its main alkaloids are currently contradictory due to the inadequate analytical methods and the wide variation in toxicity levels reported. Here, we tested the hypothesis that the ontogenetic stage of plant development may explain a significant part of the variations in the main *Equisetum*-type alkaloids. Two populations of marsh horsetail were sampled over two growing seasons. The plant material was classified according to their developmental stages and subsequently the main alkaloids were determined by hydrophilic interaction liquid chromatography and high-performance liquid chromatography electrospray tandem mass spectrometry (HILIC HPLC-ESI-MS/MS) analysis. ANOVA revealed significant effects of the ontogenetic stage but not the site on the main *Equisetum*-type alkaloids (sum of palustrine and palustridiene) ranging from 213 to 994 mg/kg dry matter (DM). The highest alkaloid content was found in the stages of early development. Not the season itself, but the growth temperature co-influenced the alkaloid content. Our results help to resolve the seemingly contradictory information provided by previous studies on the toxicity of *E. palustre* and are of practical relevance for the prevention of contamination risks in wet grassland use.

## 1. Introduction

Marsh horsetail (*Equisetum palustre* L.) is one of the 15 species of the subgenus *Equisetum* (family: *Equisetaceae*) whose members are collectively known as horsetails [1]. These pteridophytes are living remnants of an ancient group of plants dating back to the Devonian period [2]. The marsh horsetail as a circumpolar perennial species is widely distributed in wet areas of the northern hemisphere [3]. Its natural habitat consists predominantly of marshy and swampy land [4], with the peatlands being the main area of distribution. Usually, marsh horsetail grows associated with wet grassland species and is able to colonize many plant communities if the site conditions are favorable [5]. Since these wet grassland areas have been subjected to increasing pressure from the developing livestock sector [6,7] there have been reports regarding the poisoning of ruminants and horses. Weber [8] cites numerous of such evidence cases in central Europe and Scandinavia at the end of the 18th and beginning of the 19th century. From the 1950s until the mid-1970s there was again an accumulation of reports of the poisoning of farm animals due to the consumption of marsh horsetail-contaminated roughages [4,9,10,11]. During this period, the analytical methods for the determination of horsetail’s alkaloids were sparsely developed and highly variable [12,13,14,15]. It is generally assumed that a 5% content of marsh horsetail in the feed of ruminants causes severe poisoning symptoms [16]. Hence, more detailed quantitative information can serve as a prerequisite for deriving toxicological thresholds for an evidence-based risk management.

Only recently, the alkaloids of the genus *Equisetum* have been analyzed systematically by Cramer et al. [17] and in a broader taxonomical range by Tipke et al. [18] using hydrophilic interaction liquid chromatography and high-performance liquid chromatography electrospray tandem mass spectrometry (HILIC HPLC-ESI-MS/MS) resulting in reliable and reproducible analytical data. According to these data, the alkaloid profile of marsh horsetail is dominated almost exclusively by the two main piperidine alkaloids palustrine and palustridiene (Figure 1) showing concentrations ranging from 120 to 883 mg/kg dry matter (DM) (sum of both alkaloids, average 550 mg/kg DM) [18]. Another alkaloid nicotine, which is frequently mentioned in literature [17,18] as an alkaloid observed in *Equisetum* spp. occurs only at <50 µg/kg DM. This level is 3–4 orders of magnitude lower than observed for the piperidine alkaloids. Hence, a biological relevance, e.g., a deterrence effect on herbivores seems unlikely. It is more likely that these low nicotine levels are side products towards the biosynthesis of the primary metabolite NAD [18].

So far, considerable variances of the total alkaloid content were reported in literature [16,17,19]. Holz and Richter [19] reported total alkaloid concentrations of 150–3020 mg per kg DM in marsh horsetail as early as 1960. Besides the annual effects, however, the authors were unable to provide any information on the causes of this high variation. Although the standard of analysis at that time may have contributed to the alkaloid value fluctuations, it is rather unlikely that this would be the only explanation of this variation. As it is known from other poisonous plants, site and environmental conditions [20], season [21,22] and the ontogenetic stage [23,24] are likely to influence the concentrations of the toxic agents as well. Already in 1956, Uotila [25] reported that marsh horsetail collected in June was more toxic to livestock than the material collected in the autumn. However, this anecdotal finding was never proven nor causally explained by the author. Holz and Richter [19], on the other hand, assumed the opposite: an increase in alkaloid content in the course of the vegetation period. To date, there is no systematic study on marsh horsetail focusing on these seasonal interactions.

Information regarding the influence of the growth stage on the concentration of *Equisetum*-type alkaloids in horsetail infested grassland could help to avoid critical toxin concentrations in roughage by means of adapted grassland management. This aspect is particularly important as marsh horsetail is migrating back into extensively used grasslands more frequently in the course of numerous rewetting measures for the ecological restoration of formerly drained wetlands in central Europe [5,26,27,28,29].

So far, there are no studies investigating the aspect of ontogenetic development on alkaloid formation and consequently their concentrations in the biomass of *E. palustre*. Here, we address this gap of knowledge by a combination of state-of-the-art analytical methodology with the application of population-based collections of ontogenetically classified horsetail plants over a two-year period. In particular, the following questions should be addressed:(i)Will the factors such as “year” and “site/population” contribute to explain the variance of the alkaloid content?(ii)Does the ontogeny of *E. palustre* influence the alkaloid levels?(iii)Do growing conditions/season play a role in the bioformation of *Equisetum*-type alkaloids?

## 2. Results

### 2.1. Variability of the Equisetum-Type Alkaloid Contents

In this two-year study, a total of 162 cohort samples of marsh horsetail shoots originating from two different sites were analyzed for a total of 10 *Equisetum*-specific alkaloids and nicotine (Figure 1). However, only two major alkaloids, palustrine and palustridiene, were detected.

Since all other alkaloids such as nicotine were only present in trace amounts in this sample set, the sum of palustrine and palustridiene was called ‘*Equisetum*-type alkaloids’. Figure 2 shows the variability of these *Equisetum*-type alkaloids separated by survey year and sites as boxplots.

It should be noted that the observed variability in alkaloid contents was not independent of sample size. The sample size varied between sites and years according to the availability of shoots in different stages of growth and according to the grassland use patterns. For instance, the limited sample size in Rothenmoor 2019 (n = 16) was caused by drought stress in this specific year. On the other hand, the comparatively limited number of shoot collections in Parkentin in 2020 (n = 18) was a consequence of intensified grassland use including an early cutting accompanied by a reduction of marsh horsetail, which is a weak competitor under these conditions. However, the wide range of *Equisetum*-type alkaloid content (213 to 994 mg/kg DM in this study) under different environmental conditions provided the prerequisite to further analyze the causes for this variation using linear mixed effects model based analysis of variance (LMM-ANOVA).

### 2.2. Results of the Linear Mixed Model Analysis

A linear mixed model analysis was applied to quantify the effects of the ontogenetic stage of *E. palustre*, the site, the year and their interactions on the content of *Equisetum*-type alkaloids. Table 1 presents the main results of the LMM-ANOVA. A visual inspection of the residual plots did not reveal any obvious deviations from homoscedasticity or normality. Appendix A in the Appendix A shows the core model output including estimates, confidence intervals and test results.

There was a significant effect of the ontogenetic stage along with a stage by site interaction. This interaction reflected the fact that the stage-dependent alkaloid levels followed a site-specific pattern. Remarkably, neither the location itself nor the year of the survey had any directed, significant effect on the total alkaloid content. This was mainly due to the inclusion of the growth days two days before sampling as a random effect, which led to an increase in the model quality from a Marginal R^2^ of 0.34 to a Conditional R^2^ of 0.84 (see also Appendix A). The analysis of variance for the response characteristic ‘palustrine content’ showed nearly identical results (see Appendix A), because palustrine accounted for up to 92% of the total *Equisetum*-type alkaloid content and palustridiene was usually only present in minute proportions. Applying the same model structure to the response variable ‘palustridiene content’ revealed a slightly different variance pattern. Significant effects were again detected for the stage and the site as well, whereas the stage × site interaction approached significance (see Appendix A for details).

### 2.3. Effects of the Ontogenetic Stage on Main Alkaloid Contents

As the interaction stage × site was significant, it seemed appropriate to present the dependence of the alkaloid content on the growth stage separately for both collection sites (Figure 3). The stages at the x-axis are ranged from early stage (I) to full developmental stage (IV). The height of the bar columns indicates the mean of alkaloid content, the error bars correspond to the standard deviation of the means (±s_d_). The modeled estimates of the *Equisetum*-type alkaloid contents are dot-pointed.

The Parkentin site showed a higher alkaloid content. However, the general course of the stage-dependent alkaloid formation was similar for both locations despite the significant interaction of the corresponding site. The marsh horsetail plants showed a respectable alkaloid content already at a quite early growth stage when there was still only a small photosynthetically active shoot mass present. The alkaloid concentration reached its peak in development stage I.5 and was consistent for both sites although at different absolute levels. As the biomass continued to increase during the course of further ontogenetic development, the alkaloid content decreased slightly. At development stage III (developed whorls, leaf segments still growing) the minimum alkaloid concentration was observed.

Palustrine and palustridiene were detected as the two main alkaloids in the shoots of marsh horsetail. Palustridiene averaged up to 7.82% DM (range 0.0–47.1% DM) of the sum of *Equisetum*-type alkaloids and was therefore significantly less abundant than palustrine (92.18% on average with a range of 52.9–100%), which dominated the alkaloid content at all sites at all stages.

Although the averaged palustridiene percentages seemed to follow the stage dynamics of the total alkaloid contents peaking at stage I.5 and a subsequent decline (see Figure 4), their dispersion was too large and the LMM ANOVA did not reveal any significant influence of the growth stage. Furthermore, none of the environmental variables (year, location, growing degree days (GDD)) were able to influence the palustridiene percentage directly. In conclusion, both palustrine and palustridiene concentrations in above-ground shoots of *E. palustre* were similarly dependent on the growth stage (see also Appendix A in the Appendix A), but their ratio was largely unaffected from any of the investigated factors.

### 2.4. Effects of Random Variables on Main Alkaloid Contents

The growing degree days were summed up in three variants starting from 1st January of the survey year and then 20 and 2 days before the collection date, resulting in the variables GDD, GDD_20_ and GDD_2_, respectively. Only GDD_2_ (used as a random variable in the LMM) contributed significantly to the explanation of the observed alkaloid content (see Table 1). 

As shown in Figure 5, the cumulative growth temperatures on GDD_2_ at the two sites had different effects on the total alkaloid content. At the extensively managed Rothenmoor site, with largely undisturbed vegetational development, an increase in GDD_2_ resulted in a more continuous increase in the alkaloid contents. In contrast, at the more intensively used Parkentin site, the growth temperatures only increased the alkaloid contents up to a GDD_2_ of approximately 18, shifting to an opposite effect afterwards around midsummer.

The average shoot length was not included in the linear mixed models because of their naturally strong intercorrelation with the growth stages (Kendall’s τ for growth stage versus shoot length = 0.78, z = 13.58, *p* < 0.001 ***). Nevertheless, we used linear regression analysis to investigate any influence of shoot length on the alkaloid contents at the level of the individual developmental stages. However, the variation of the average shoot lengths of a stage-sorted collection cohort could not explain the differences in alkaloid concentration in any of the growth stages. We therefore decided to refrain from presenting the non-directional scatterplots here.

## 3. Discussion

### 3.1. Variation of Equisetum-Type Alkaloid Levels

Despite the fact that only two sites were sampled over two vegetation periods, a wide range of alkaloid concentration (781 mg/kg DM) was found. This result confirmed the observations of the past by other authors researching the toxicity of marsh horsetail [17,19]. Only recently, Tipke et al. [18], who also determined the *Equisetum*-type alkaloids using the HILIC HPLC-ESI-MS/MS method, reported a similar range of total *Equisetum*-type alkaloids (ranging from 120 to 883 mg/kg DM) in samples taken from 10 different sites. However, the interpretation of such site effects is difficult as long as there is no genetic information for population characterization combined with reliable environmental variables that would help to resolve the genotype × environmental interaction. In this two-site study, a similar high variability of alkaloid contents was observed as in a global multi-site collection with much larger site amplitudes [18]. This observation suggests that the location or the site-bound population was not the main cause of variation in alkaloid content.

### 3.2. Effect of the Ontogenetic Development on Equisetum-Type Alkaloid Contents 

This present study demonstrates for the first time that the alkaloid content of marsh horsetail depends on its stage of development. This approach, which is quite obvious from the point of view of both plant defense ecology [30] and plant metabolism [31], has so far only been followed for some other poisonous plants [24]. The highest alkaloid content was found in the stage of early development of lateral leaf branches of *E. palustre*. At this very early stage of development, this demands a lot of effort from the plant metabolism due to a trade-off between growth rate and energy intensive alkaloid production [32]. It can therefore be assumed that such early protection against herbivores was necessary during the evolution of this species [33]. The drop in alkaloid levels to stage III (developed whorls) might be due to a subsequent dilution effect caused by the enhanced formation of a cell wall substance. It can be suspected that (a) at a later stage of development, the already advanced silicate storage in *Equisetum* [34] takes over part of the herbivorous protection, hence additional energy costs for alkaloid production is not required and/or (b) that the fully developed plant will no longer be overlooked by large grazing animals thus avoiding defoliation. This would be in accordance with observations that cattle in particular avoid the uptake of forage with detectable amounts of marsh horsetail [25,35].

We have found that both palustrine and palustridiene concentrations in the shoots of marsh horsetail show a similar pattern in the course of ontogenetic development but their ratio to each other remains largely constant. These data point to a close, genetically determined and metabolically based dependency of these two compounds. Although the biosynthesis of the piperidine alkaloids in *Equisetum* spp. is not elucidated by experimental studies yet, a mixed biosynthetic pathway via the condensation of spermidine and a C10-fragment (derived from β-oxidative degradation of fatty acids) can be assumed [17]. Furthermore, based on the structures (Figure 1), it can be assumed that palustridiene is the biosynthetically precursor of palustrine which represents a biosynthetic sink and so far, is the only *Equisetum*-type alkaloid whose toxicity to mammals is confirmed. Interestingly, *E. palustre* is the only species in this ancient group of spore-producing plants that produces significant amounts of alkaloids while closely related species totally lack these secondary metabolites [17]. However, further research is necessary to evaluate the ecological role of these plant alkaloids.

### 3.3. The Ambiguous Role of Season on Alkaloid Levels and Toxicity

An interesting side aspect of the analysis was the finding that the inclusion of growing degree days two days before sampling (GDD_2_) in the linear mixed effects model led to a better explanation of *Equisetum*-type alkaloid contents while the whole year GDD did not. The latter could be regarded as a good proxy for a season, which so far in literature, has been considered to be the main parameter influencing palustrine concentration [16,19,25]. In contrast, the parameter GDD_2_ reflected the weather conditions of the growth phase immediately before sampling. In literature, an increase in the alkaloid content with rising temperature was also confirmed for *Lupinus angustifolius* [36] and endophyte infected *Lolium* [37]. However, as temperature and global radiation are correlated during the growing season [38], the effect of GDD_2_ could also be due to an increased photosynthesis performance.

The apparently contradictory findings in literature on the seasonality of the toxic effect of *E. palustre* could be explained by the fact that different stages of development under different weather conditions were compared, hence the season was only erroneously used as a sample characterization feature. Additionally, the season should be in general an unsuitable criterion for estimating the toxicity of marsh horsetail because *E. palustre* produces new shoots from the rhizomes until late summer. Unlike, for example, in shrubs, there is conclusively no congruence between ontogenetic development and seasonal weather patterns.

### 3.4. Aspects of Reducing the Risk of Feed Contamination

The results on the stage-dependence of alkaloid concentrations of marsh horsetail have quite a practical relevance and may help to minimize the contamination risks for farm animals. Early grazing practices, as traditionally applied on the brackish marshes along the North Sea coast to promote sward density and to control the early heading meadow foxtail, should be avoided when cattle pastures are infested by marsh horsetail. Despite lower alkaloid concentrations in later development stages, the naturally more pronounced biomass may still lead to higher alkaloid quantities per unit grassland area. However, because the grass-dominated accompanying vegetation increases in biomass in the same way and marsh horsetail does not outcompete grasses, the absolute increase in alkaloid quantity does not lead to an increase in palustrine concentrations in the feedstock as long as the grasses are vital. Therefore, all measures that promote valuable grasses on wet grassland sites are the most effective way to reduce feed contamination with *Equisetum*-type alkaloids [39].

## 4. Conclusions

We conclude that the considerable variation in *Equisetum*-type alkaloid contents of marsh horsetail can be explained to a significant extent by the ontogenetic stage of development. The already high alkaloid levels in the early developmental phase can be regarded as a protection mechanism against herbivores at a time when the energy content of the plants would be attractive to herbivores. Early grazing practices on marsh horsetail infested sites should be avoided. The common practice that a 5% coverage of marsh horsetail at wet grassland swards is regarded as a threshold that should not be exceeded is considered too high for contemporary, performance-based early grassland use with the ambition of a high animal health status.

## 5. Materials and Methods

### 5.1. Collection Sites

Plant material was collected from two *Equisetum palustre* populations at wet grassland sites in Parkentin (54°04′52.9″ N; 11°57′23.9″ E) and in Rothenmoor (53°40′29.4″ N; 12°37′51.7″ E) over the vegetation periods 2019 and 2020. Locations of the collection sites are shown in Appendix A and the embedding of the sites into the landscape is shown in Appendix A.

At both sites, mineral sediments (loamy sand) in the underground covered by a shallow, degraded peat layer (<50 cm) characterize the dominating soil conditions. Both locations exhibit a high degree of soil moisture in the winter half of the year but falling dry in the middle of summer. While the soil moisture of the Parkentin site is caused by high groundwater levels, subsurface percolation water from the adjacent moraine slope is mainly responsible for the wet conditions in Rothenmoor. The accompanied vegetation of the marsh horsetail on both sites is a sedge-rich plant association of the *Angelico sylvestris-Cirsietum oleracea*-type. *Festuca pratensis*, *Agrostis* ssp., *Holcus lanatus*, *Deschampsia cespitosa*, *Ranunculus acris*, *Carex riparia* and *Carex nigra* are frequently occurring species at both sites. While the three-cut meadow Parkentin is used regularly, Rothenmoor is an extensive meadow with an irregular pattern of use (1–2 cuts per year).

Both years were characterized by mild and wet winters followed by a dry spring with late frosts. The weather pattern during the entire survey period is shown as a climate diagram according to Guijarro [40] in the Appendix A (Appendix A). Both sites were chosen to mirror a climatic gradient with one site representing a maritime coastal location (Parkentin) to a more continental weather pattern with stronger temperature fluctuations inland (Rothenmoor).

### 5.2. Sampling Procedure and Sample Preparations

The plants were cut immediately above the soil surface and transported in cooler bags for further classification and preparation. Plant material was classified into four phenologically defined main groups of vegetative sprouts (Figure 6) immediately after collection. The procedure is given in Appendix A of the Appendix A.

Classification scheme:(I)Early stage, main axis without visible lateral branches;(II)Lateral leaf branches visible, beginning of whorled phyllotaxis;(III)Developed whorls, leaf segments still growing;(IV)Fully developed growth stage, leaf segments are expanded.

For a more precise capture of the phenological developmental process, the intermediate categories I.5 (between I and II) and II.5 (between II and III) were created in addition to the four main developmental stages. It should be noted that the different stages of growth did not occur in strict chronological order. Plants in early vegetative developmental stages were still present at later collection dates. These circumstances carry the risk that any seasonal effect on the expression of alkaloid contents is not evenly and randomly distributed within the phenological classes. However, this was considered during the course of data analysis by including the growing degree days and the average plant height as an additional random variable.

The stage classified material was oven-dried to a constant weight at 45 °C and then ground to powder using a laboratory rotary mill (Brabender, Duisburg, Germany). Powders were stored at room temperature in the dark until sample extraction.

### 5.3. Determination of Covariables

With the growing degree days (GDD) and the average shoot length per collection cohort (SL_C_) two more covariables were considered that reflected the intensity of horsetail growth. GDD were calculated as follows:(1)$$\mathrm{GDD}\;=\;\frac{\mathrm{Tmax}-\mathrm{Tmin}}{2}-\mathrm{Tbase}.$$

Equation (1) uses the average of the daily maximum (Tmax) and minimum (Tmin) temperatures compared with a base temperature *T*base at which the plant grows. We estimated a base temperature of 6 °C for marsh horsetail. GDDs were calculated as the sum from 1 January of the survey year (GDD) and as the sum of 20 days (GDD_20_) and 2 days (GDD_2_) before sampling for each sampling event using data from the nearest weather station of the German Weather Service (DWD).

The average shoot length was calculated as the mean of the single plant shoot length per stage cohort of each sampling event. The shoot lengths of the single plants of a cohort (sampling date × site × stage) were measured with a ruler during the classification process (see Appendix A).

### 5.4. Alkaloid Analysis

HILIC HPLC-MS/MS in ESI positive mode was applied to analyze and quantify the alkaloid content of the *Equisetum* samples.

#### 5.4.1. Chemicals and Reagents

The used solvents, methanol and acetonitrile, were of LC-MS grade quality. Modifiers for the LC-MS solvents, formic acid (≥98% puriss., p.a.) and ammonium acetate (traceSelect), were obtained from Honeywell/Riedel-del-Haen (Morristown, NJ, USA). Water was double distilled before use.

#### 5.4.2. Extraction and Sample Preparation

Approximately 50 mg of accurately weighted plant material (duplicates per sample) were suspended in 1.7 mL of 80% methanol in 2 mL Eppendorf tubes (Safe-Lock, Eppendorf, Hamburg, Germany) and extracted using 15 min ultrasound treatment in sweep mode (Elmasonic S120, 12.75 L, 37 KHz, effective ultrasound power 200 W; Elma Hans Schmidbauer GmbH & Co. KG, Singen, Germany). Afterwards, the samples were centrifuged (13,400 rpm, 7 min, Minispin, Eppendorf, Hamburg, Germany). The supernatant was filtered through cotton wool into 4 mL vials (N13, Macherey-Nagel, Düren, Germany). The extraction was repeated three more times using 1.3 mL 80% methanol each. The extracts were combined and concentrated to dryness at 65 °C using a constant stream of pressurized air. The residue was re-suspended using 2 mL of LC-MS grade methanol and solubilization was improved by applying ultrasound treatment for 15 min. A two-step dilution was performed in 1 mL of each (first: 1/100 in LC-MS grade methanol followed by 1/50 dilution in 70% acetonitrile) to yield a final dilution of the extract of 1/5000, which was directly used for the chemical analysis.

#### 5.4.3. HPLC-ESI-MS/MS Analysis

A recently established method, the use of HILIC chromatographic separation and ESI-MS/MS detection in the multiple-reaction-monitoring mode (MRM), was used for the detection and quantification of all so far known *Equisetum* alkaloids [18]. The method comprised the detection of 10 *Equisetum*-type alkaloids and nicotine. Briefly, the diluted extracts were analyzed using a 1200 Series HPLC (Agilent, Waldbronn, Germany) coupled to a 3200 QTrap mass spectrometer (Applied Biosystem/MDS SCIEX, Darmstadt, Germany) applying ESI-MS/MS detection in the multiple-reaction-monitoring mode (MRM). A 100 × 2.1 mm (1.7 µm) Poroshell 120 HILIC column (Agilent, Waldbronn, Germany) was used for separation. The mobile phase consisted of 30 mM ammonium acetate/0.2% formic acid solution (solvent A) and 0.3% formic acid in acetonitrile (solvent B). The following gradient was used at a flow rate of 400 µL/min: 0–2 min (5% A), 2–18 min (5–95% A), 18–21 min (95% A), 21–22 min (95–5% A) and 22–32 min (re-equilibration 5% A). All details concerning the applied MS parameters and MS/MS transitions were added to the electronic Appendix A.

Analyst 1.6.2 Software (Applied Biosystems MDS Sciex, Darmstadt, Germany) was used for data analysis and area integration of every target compound peak.

Quantification of palustrine and palustridiene was achieved using an external calibration method using authentic reference material [17]. Calibration curves for palustrine and palustridiene were recorded before and after each set of 20 samples using calibration solutions at 0.625, 1.25, 2.5, 5.0, 10.0, 20.0 and 40 ng/mL and were shown to be linear in this concentration range and all samples analyzed were within the calibrated range. Most recent calibration curves were generated in Microsoft Excel^©^ for each sample set and were used to calculate the alkaloid content of each sample. limit of detection (LoD; 1 mg/kg) and limit of quantification (LoQ; 5 mg/kg) were determined at a signal-to-noise ratio of 3:1 and 10:1, respectively.

### 5.5. Data Analysis

We used a linear mixed model (LMM) analysis in R (version 3.6.1) [41] to model main alkaloid contents as a function of ontogenetic stage, site, year and optional covariables characterizing growth rate. We performed LMM with the alkaloid contents as the dependent variable and set ‘stage’ and ‘site’ as well as their interaction as fixed effects specifying them by the lme4 package [42]. ‘Year’ and ‘growing degree days’ were included as random effects. The significance was calculated using the lmerTest package [43], which applies Satterthwaite’s method to estimate degrees of freedom and generate p-values for mixed models. The model specification was as follows: alkaloid content ~ stage + site + stage × site + (1|year) + (1|GDD_2_). Significant differences of the estimated means were post-hoc tested by t-tests with Tukey adjustment and Kenward–Roger approximation for the degrees of freedom. We used Kendall’s tau (τ) to correlate shoot length with stages and local weighted regression (*loess*) to figure out the relationship between GDD_2_ and alkaloid contents.

## Figures and Tables

**Figure 1 toxins-12-00710-f001:**
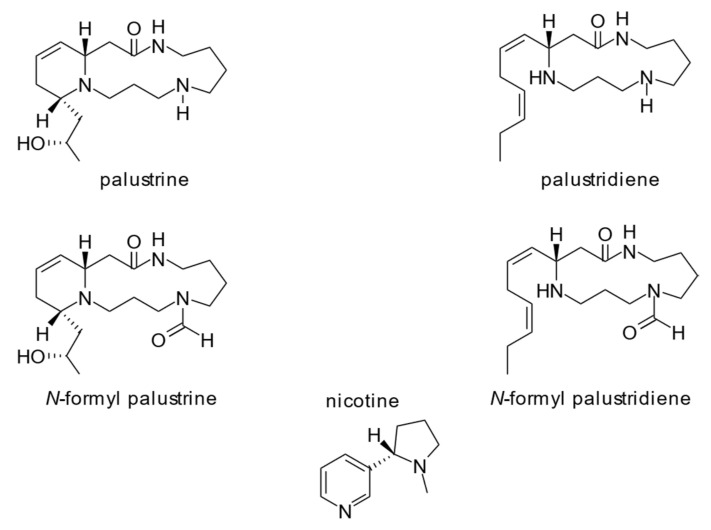
Structures of the four major *Equisetum*-type alkaloids of *E. palustre*, including nicotine, according to Tipke et al. (2019) [18].

**Figure 2 toxins-12-00710-f002:**
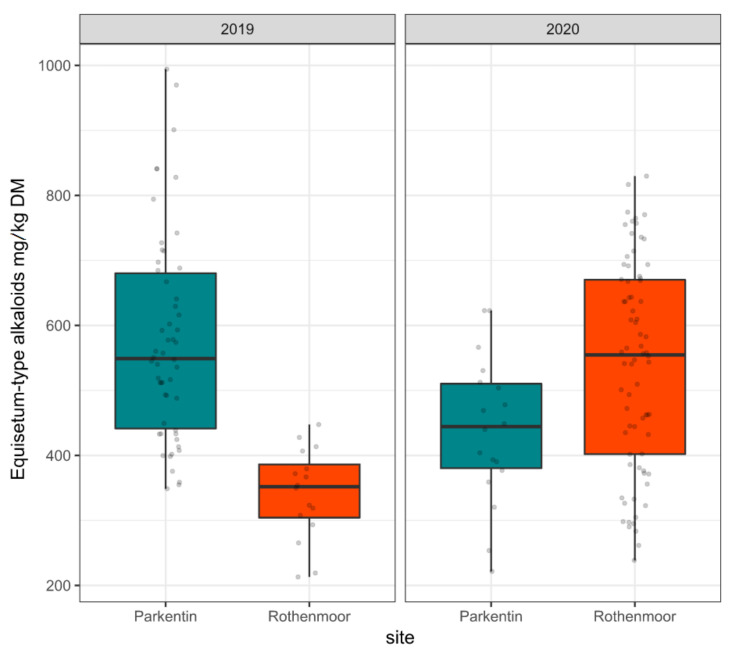
Variability of the *Equisetum*-type alkaloid contents (sum of palustrine and palustridiene) presented as boxplots separated by survey year and collection site. Raw data are dot-plotted. The boxes are bounded on the top by the third quartile and on the bottom by the first quartile; the median divides the box; the whiskers represent observations that are 1.5 time greater than the interquartile range.

**Figure 3 toxins-12-00710-f003:**
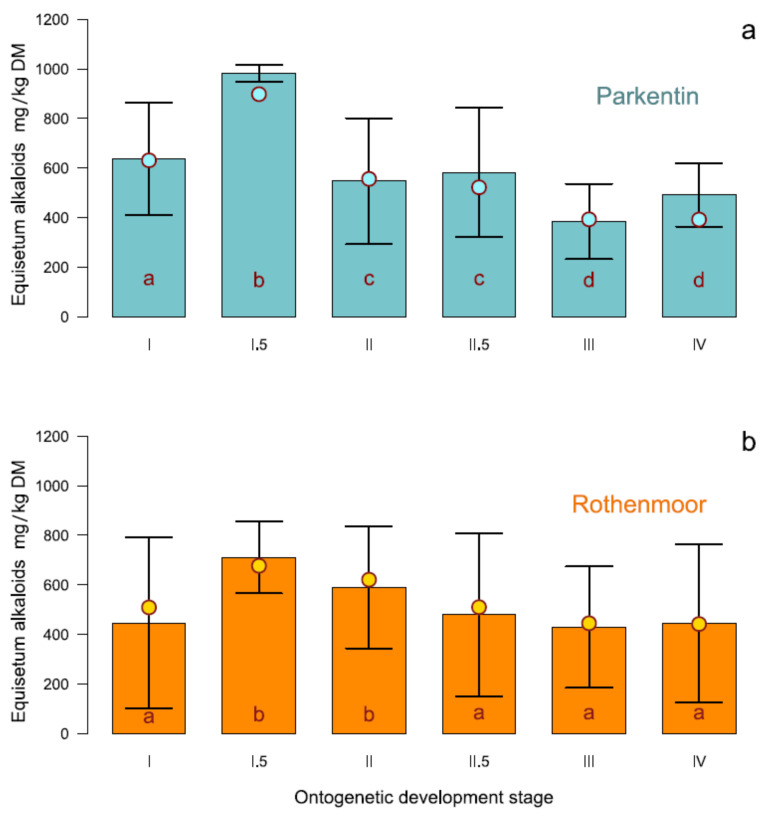
Concentrations of *Equisetum*-type alkaloids in the shoots of *E. palustre* L. in relation to the ontogenetic stage of plant development. (**a**) Parkentin site; (**b**) Rothenmoor site. The LMM modeled means are dot-pointed. Error bars = standard deviation of the means (±s_d_). Different letters indicate significant differences of the estimated stage means at site level (Tukey adjustment of *t*-test, Kenward–Roger approximation for DF, *p* < 0.05).

**Figure 4 toxins-12-00710-f004:**
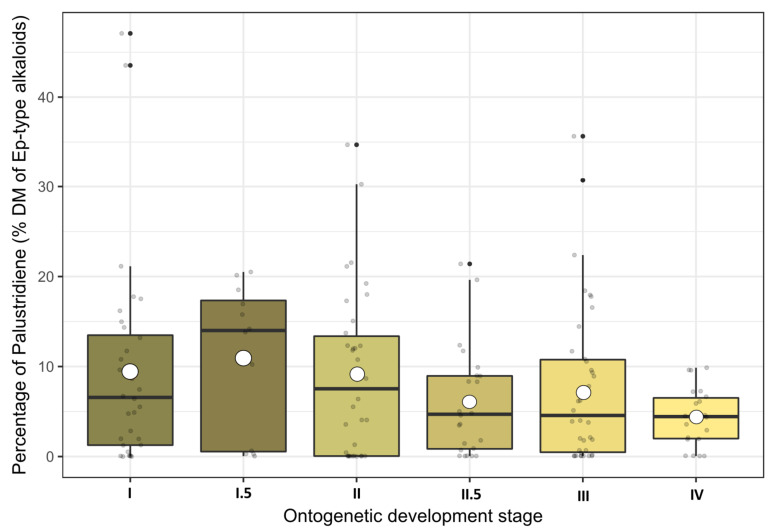
Percentages of palustridiene contents presented as boxplots and arranged according to their ontogenetic development stages. Raw data are dot-plotted. The boxes are bounded on the top by the third quartile and on the bottom by the first quartile; the median divides the box; the whiskers represent observations that are 1.5 times greater than the interquartile range. White dots inside the boxes indicate the stage means.

**Figure 5 toxins-12-00710-f005:**
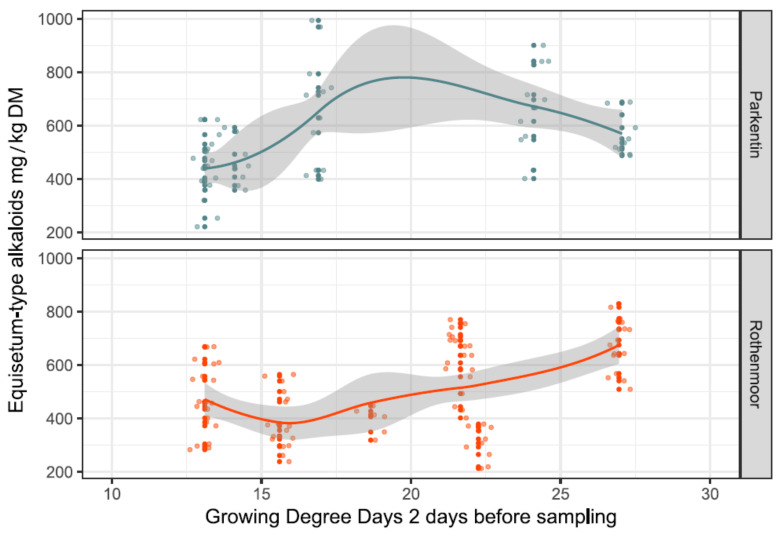
Site-specific influence of the growing degree days two days before the sampling events (GDD_2_) on the concentrations of *Equisetum*-type alkaloids in the sampled shoots of *E. palustre.* The raw data are dot-plotted. Trendlines were constructed using local regression functions; the corridors (grey) around the trendlines indicate the confidence intervals.

**Figure 6 toxins-12-00710-f006:**
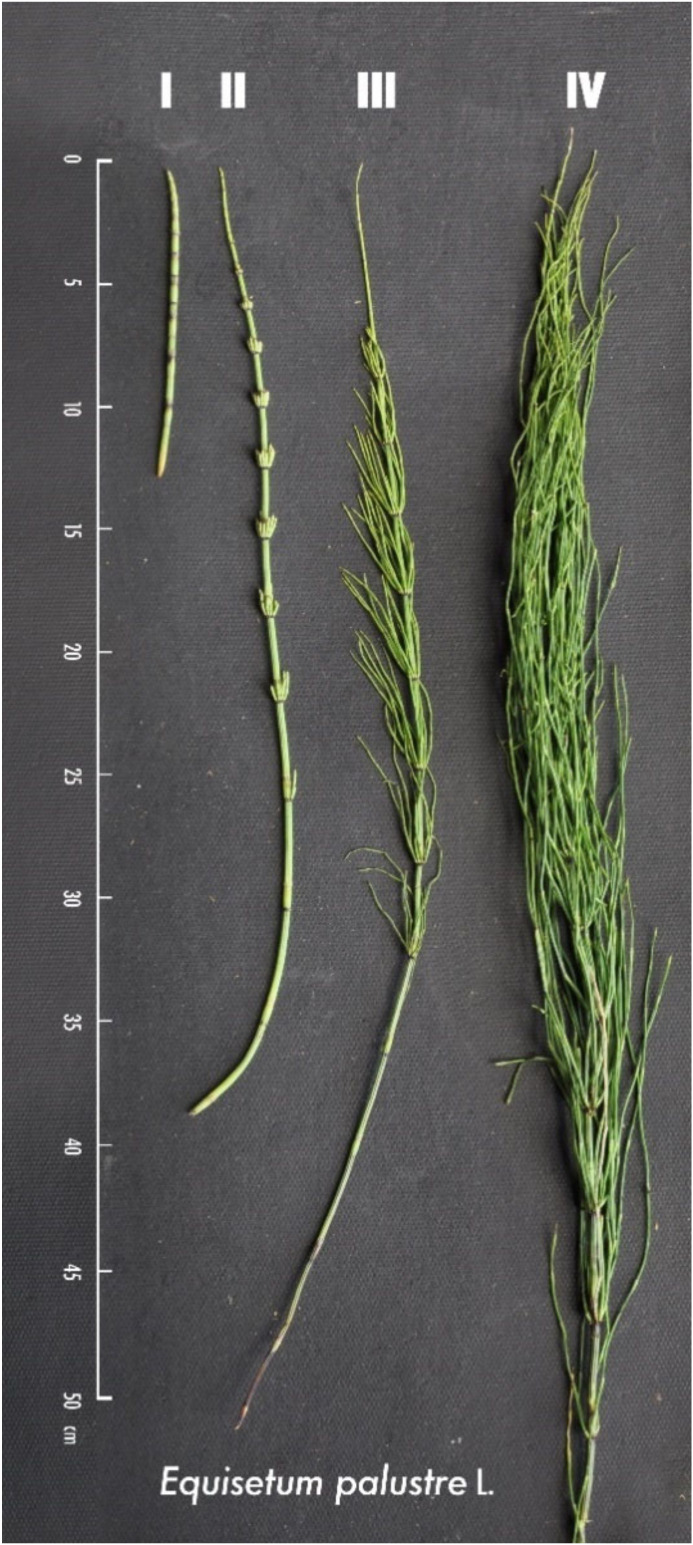
Classification of vegetative sprouts of *Equisetum palustre* L. into four main ontogenetical stages illustrated with typical single plant specimens.

**Table 1 toxins-12-00710-t001:** Results of the linear mixed effects model based analysis of variance (LMM-ANOVA) with Satterthwaite’s method to test the effects of ontogenetic stage, site and the interaction of stage and site as fixed variables and the effects of year and Growing Degree Days accumulated over two days before sampling (GDD_2_) as random variables on the *Equisetum*-type alkaloid concentrations in shoots of marsh horsetail.

Fixed Effect	NDF 1	DDF 2	*F*-Value	*p*
Ontogenetic stage	5	140.82	53.51	<0.001 ***
Sampling site	1	149.53	1.64	0.203 n.s.
Interaction stage × site	5	140.91	11.17	<0.001 ***
**Random Effect**	**DF 3**	**Chi-squared**		***p***
Year of survey	1	0.00		n.s.
Growing degree days 2	1	155.97		<0.001 ***

^1^ Nominator Degrees of Freedom (NDF), ^2^ Denominator Degrees of Freedom (DDF), ^3^ Degrees of Freedom (DF), not significant (n.s.). (***) *p* < 0.001.

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
