# Peer review of "Variation of the Main Alkaloid Content in Equisetum palustre L. in the Light of Its Ontogeny"

_toxins, 2020, doi:10.3390/toxins12110710_

Round 1

Reviewer 1 Report

General:

I consider that according to the presented results the variation of the total alkaloid content in marsh horsetail might be related to ontogenetic stage of plant development. Overall the paper is well written and presented. However, I do not see any supplementary materials to fully assess the results not included in the main body of MS.

In my opinion the title, could be revised to better reflect the main objectives. For that purpose the authors could consider to change the wording “main alkaloid contents” into i.e. “variation of the alkaloid content” or “total alkaloid content”.

Specific comments for the authors:

Introduction:

Additional information should be incorporated to facilitate further MS reading:

  • please include the information that palustrine and palustridine are the major/main (unified naming throughout the MS) alkaloids of the genus and typically no other alkaloids are detected. If otherwise please describe. Do they belong to piperidine based alkaloids?
  • Why nicotine is also referred to in the results section? Could it be also detected in other Equisetum species?
  • Is there anything known about the biosynthetic pathway of the alkaloids in Equisetum?

L25: there is no need to use italics for the name of plant family

L41: genus name – italic

L46: DM abbreviation should be explained at first use

L47: “In view of the known toxicity of palustrine… “ – uncear sentence, please consider rewording.

L65-66: unclear sentence; “extensively used”?; expression “more frequently” is unnecessary?

Results:

L120: The analysis of variance for the response characteristic ‘palustrine content’ showed nearly identical results – where is this data shown?

L131-141: please consider lettering included in Figure 3 indicating significant differences of the estimated mean alkaloid content in particular stages. The differences in the alkaloid concentration across ontogenetic stages in Rothenmoor site are not spectacular so it is difficult to indicate minimum alkaloid concentration in stage III as it is statistically not different from stage IIb and IV. This also applies to section 3.2 of the discussion.

L148: the content and range of the palustridiene is presented in the third subsection of the Results, while it would be better to know this information from the beginning. Please include the same information for the palustrine.

Discussion

L190-191: „…a wide range of alkaloid concentration (781 mg kg DM-1) was found.” Why is it different from 101: “However, the wide range of Equisetum-type alkaloid content (213 to 994 mg kg DM-1 in this study)…”

L197: very long sentence, please consider re-wording or shortening. i.e.:

The finding of this study showing that similarly high variability of alkaloid contents was observed for two locations, as previously demonstrated for a multi-site collection [18] suggests, that the location, or the site-bound population was not the main cause of variation.

L217: “We have found that both palustrine and palustridiene concentrations in the shoots of marsh horsetail show a similar pattern in the course of ontogenetic development…” – I do not see the pattern of palustrine shown in the main text nor reference to supplementary data, so it is difficult to assess if the ratio is constant.

L220: “…It can be assumed that palustridiene is the biosynthetically precursor of palustrine, representing..” – Consistent ratio of both alkaloids or determined abundance are not sufficient arguments for such assumption. For instance in narrow-leafed lupin tetracyclic lupanine is the most abundant alkaloid and it is further converted to angustifoline and 13OH-lupanine the other two major alkaloid of the species. Tetracyclic multiflorine is only occasionally detected for this species while for other lupins it might be a major alkaloid together with bicyclic lupinine which are probably synthesized independently.

Are there any other premises to assume which of the two alkaloids detected for E. palustre is the precursor alkaloid i.e. arising from chemical structure?

L232-233: the reference for Taraxacum officinale [36] is not the most accurate as it covers the roots and secondary metabolites in general, with no specific analysis of the alkaloids.

L 253: “that” should be omitted

Conclusions:

L 261-264: the last sentence in the conclusion section shall be reworded  or shortened.

Materials and Methods

L286: Number of samples collected in each site and year should be listed in material and methods.

L322: Figure A3 not included – is this supplementary marerial?

L331: it should be specified that 50mg applies to dry plant material, and should be also explained if it was further divided into two technical replicates, or 50mg was needed for each technical replicate.

Author Response

We are grateful that you spent your valuable time in working through the manuscript. The input you have given and your contributions definitely helped to improve our manuscript. We have modified the manuscript in response to the insightful comments. For your convenience, we marked and discussed our modifications in the attached response document.

We look forward to hearing from you regarding our submission. We would be glad to respond to any further questions and comments that you might have.

Kind regards,

Jürgen Müller, Philipp Puttich and Till Beuerle

Reviewer 2 Report

The Authors have investigated an interesting topic related to plant toxicology and possible health risks, and the theme has been properly described. I would like to congratulate authors for the good-quality of the article, the literature reported used to write the paper, and for the clear and appropriate structure. The manuscript is well written, presented and discussed, and understandable to a specialist readership.

In general, the organization and the structure of the article are satisfactory and in agreement with the journal instructions for authors. The subject is adequate with the overall journal scope. The work shows a conscientious study in which a very exhaustive discussion of the literature available has been carried out. The introduction provides sufficient background, and the other sections include results clearly presented and analyzed exhaustively.

So, I recommend the acceptance of the paper in Toxins.

Author Response

We are grateful that you spent your valuable time in working through the manuscript. Thank you for the appreciation you have shown for our contribution. Should any questions arise regarding the newly submitted version, we will be happy to answer them.